# Association of the risk of obstructive sleep apnoea with the severity of COVID-19

**Nissim Arish**[1]*, **Gabriel Izbicki**[1], **Ariel Rokach**[1], **Amir Jarjou'i**[1], **George Kalak**[1], **Shmuel Goldberg**[2]

1 Respiratory Research Unit, Pulmonary Institute, Shaare Zedek Medical Center and Faculty of Medicine, Hebrew University of Jerusalem, Jerusalem, Israel, 2 Pediatric Unit, Pulmonary Institute, Shaare Zedek Medical Center and Faculty of Medicine, Hebrew University of Jerusalem, Jerusalem, Israel

* narish@szmc.org.il

## Abstract

Patients with coronavirus 2019 (COVID-19) and obstructive sleep apnoea (OSA) have a worse prognosis than COVID-19 patients without OSA. This study aimed to examine the relationship between OSA risk and the severity of COVID-19 in patients undiagnosed with OSA. Patients diagnosed with COVID-19 and hospitalized or admitted to a community hotel were recruited for the study after recovery during a clinic check-up visit 6–8 weeks after discharge. At this visit, they answered the Epworth Sleeping Scale (ESS) and Berlin questionnaire. Demographic and clinical details were collected from electronic medical records. OSA risk was observed in 37 of 119 included patients (31.1%). Patients with high OSA risk were male, significantly older, had a higher body mass index (BMI), and had higher rates of hypertension and snoring than patients with low OSA risk. Moreover, OSA risk was associated with COVID-19 severity; 48.6% of patients with high risk for OSA suffered from severe COVID-19 compared to 22% of patients with low risk for OSA (p = 0.007). The duration of hospitalization for patients with a high OSA risk was 10.97±9.43 days, while that for those with a low OSA risk was 4.71±6.86 days (p = 0.001). After adjusting for BMI, age, hypertension, and chronic disease, the odds ratio was 4.3 (95%CI, 1.2–16, p = 0.029). A high OSA risk was associated with severe COVID-19 and longer hospitalization. Thus, we recommend that the Berlin and ESS questionnaires be completed for every COVID-19-infected patient at hospitalization, especially in the presence of comorbidities.

## Introduction

Obstructive sleep apnoea (OSA) is characterized by a repetitive partial or complete airway blockage during sleep, leading to interruptions in breathing, blood oxygen desaturation, and arousal. Consequently, people with OSA often experience severe daytime drowsiness, fatigue, irritability, and difficulty concentrating and may have a higher risk of work-related accidents. Although OSA is not considered an immediate life-threatening condition, its consequences can be severe, particularly in the presence of specific comorbidities. OSA is associated with an increased prevalence of hypertension (39%), obesity (34%), depression (19%), diabetes mellitus (15%), and asthma (4%) [1, 2].

**Data Availability Statement:** data are available from OSF DOI 10.17605/OSF.IO/7VUTX.

**Funding:** The authors received no specific funding for this work.

**Competing interests:** The authors have declared that no competing interests exist.

Due to the high transmission rate of the acute respiratory syndrome coronavirus 2 (SARS-CoV-2/COVID-19), a pandemic was declared in March 2020 [3]. COVID-19 can cause fever, cough, fatigue, and anosmia. It is also manifests as severe life-threatening conditions, such as respiratory distress, arrhythmia, sepsis, shock, loss of consciousness, and death [4, 5]. Comorbidities such as cardiovascular disease, diabetes, hypertension, obesity, chronic lung or kidney disease, dyslipidemia, and tobacco use are associated with more severe manifestations of COVID-19 and increased mortality [6–11].

Obesity, male sex, and older age are associated with severe COVID-19 and are risk factors for OSA [4, 5]. In addition, comorbidities such as diabetes mellitus, cardiovascular disease, and hypertension were found to be risk factors for poor prognosis in COVID-19 and OSA [4, 12, 13].

A higher prevalence of OSA has been observed in patients hospitalized with COVID-19 [14]. Recent studies have suggested that individuals with OSA may be predisposed to COVID-19 [15]. Maas et al. screened 9,405 cases of COVID-19 infections from 10 hospitals in the Chicago metropolitan area. They observed that OSA patients experienced an approximately 8-fold higher risk of COVID-19 compared to a similarly aged non-OSA population [14]. Recent reports have shown that patients with COVID-19 and OSA have poorer prognoses compared to COVID-19 patients without OSA, with an increased risk of hospitalization and death [14, 16, 17].

The suggested mechanisms by which OSA increases the risk of poor outcomes from COVID-19 include exacerbation or endothelial dysfunction, inflammation, oxidative stress, microaspiration, and lung injury [18–23]. Comorbidities such as hypertension, heart failure, coronary artery disease, cerebrovascular diseases, diabetes mellitus, and obesity, common among OSA patients, are also risk factors for mortality in COVID-19 patients [24]. In addition, many COVID-19 patients suffer from pulmonary fibrosis, which is a risk factor for the future development of OSA [25].

This study aimed to examine the relationship between the risk of OSA and the severity of COVID-19 in patients not diagnosed with OSA.

## Methods

### Study design

This retrospective, uncontrolled cohort study was conducted at the COVID-19 post-acute clinic at the Pulmonary Institute, Shaare Zedek Medical Center, Jerusalem, Israel. Patients diagnosed with COVID-19 who were hospitalized or admitted to a community hotel setting (hotels that hosted patients diagnosed with COVID-19 but did not need to be hospitalized) and patients who stayed in these settings until their COVID-19 test was negative) were included in the study. The patients were recruited to the study during a follow-up visit at the clinic, 6–8 weeks after discharge, between June and November 2020.

The "Helsinki Ethics Committee" of the hospital approved this study [Reg: 0174-20-SZMC]. All the participants provided written informed consent.

### Data collection

The patients' electronic medical records were reviewed to collect demographic and clinical details, including age, body mass index (BMI), sex, the time between the onset of symptoms and arrival at the emergency room, duration of hospitalization, smoking history, comorbidities (hypertension, type 2 diabetes mellitus, pulmonary and cardiovascular liver disease, and other chronic diseases and immunodeficiency), and chronic medication.

All patients were asked to answer the Epworth Sleeping Scale (ESS) and Berlin questionnaire. The ESS is a self-administered questionnaire comprising eight questions. On a 4-point scale (0–3), respondents were asked to rate their usual chances of dozing off or falling asleep while engaged in eight different activities. Daytime sleepiness was classified as 0–5 Lower Normal; 6–10 Higher Normal; 11–12 Mild Excessive, 13–15 Moderate Excessive; 16–24 Severe Excessive (https://epworthsleepinessscale.com/about-the-ess/). The Berlin questionnaire is a survey used to screen patients with OSA. It consists of 10 questions in three categories related to the presence and severity of snoring, frequency of daytime sleepiness, and the presence of obesity or hypertension. The category of snoring was positive if the total score was $\geq 2$ points. The category of fatigue and falling asleep was positive if the total score was $\geq 2$ points. Category 3 was positive if the answer to item 10 was 'Yes' or if the BMI of the patient was greater than 30 kg/m$^2$. High risk was defined if there were two or more categories where the score was positive, whereas low risk was defined if there was only one or no category where the score was positive.,

## Disease diagnosis and severity evaluation

In all cases, COVID-19 diagnosis was based on real-time polymerase chain reaction testing of samples obtained using oropharyngeal or nasopharyngeal swabs from subjects with suspected disease (e.g., symptoms, exposure, abnormal imaging, and laboratory tests).

Disease severity was evaluated during hospitalization or admission to the community hotel setting. Patients were classified into three groups: mild (individuals presenting symptoms of COVID-19, such as fever, cough, weakness, and the loss of taste and smell); moderate (individuals presenting clinical or radiographic diagnosis of COVID-19 pneumonia); severe (individuals with COVID-19 diagnosis and one of the following criteria: breaths of over 30 per minute, blood oxygen saturation of 93% or less without oxygen support, and PaO2/FiO2 rate lower than 300); and respiratory/critical) patients in need of invasive or non-invasive respiratory mechanical support) or severe damage to systemic function (shock, heart, liver damage, or kidney damage). These groups corresponded to those reported by the General Medicine Division of the Israeli Ministry of Health, published on July 12, 2020. In the data analysis of this study, mild and moderate diseases were defined as mild, whereas severe and respiratory/critical diseases were defined as severe.

## Statistical analysis

Means and standard deviations were calculated for all ratio variables, and t-tests were carried out to compare morbidity levels (high/low). For all nominal variables, absolute frequencies and percentages were calculated, and the Chi-square test with Yates's correction was carried out to compare morbidity levels (high/low). Logistic regression analysis was performed to examine the degree of connection between the risk of respiratory arrest and snoring with morbidity level (high/low).

Statistical analyses were conducted using SPSS software (version 21). The criterion for significance was alpha ($\alpha$) = .05 (two-sided).

## Results

### Characterizations of COVID-19 patients

Data from 119 patients were analyzed. The mean age of the patients was 51.58±16.3 (range, 19–83 years), BMI 27.87±8.73 (range, 17.60–54.70 kg/m$^2$), and there was an equal percentage of men and women. Most patients (69.7%) presented with mild COVID-19, while the rest

(31.3%) suffered from severe disease. The mean duration of hospitalization was 6.62±8.22 (range, 0–37 days). Only 2.5% of patients reported being smokers. Regarding comorbidities, 22.1% had hypertension, and 18.2% had diabetes mellitus. Other chronic diseases were present in 40.2% of the patients, and 48.5% regularly used various medications. Half of the patients (51.5%) reported that they snored, and 31.1% were at risk for OSA. The mean sleeping hours per night was 6.22±1.5 (range, 2–12 hours), and the mean ESS scale was 7.6±5.37 (range, 0–21) (Table 1).

## Association of the various characteristics of COVID-19 patients with the severity of the disease

Most of the patients who presented with severe disease were men, while most women were diagnosed with mild disease (male: 44.7% with mild disease vs. 63.9% with severe disease, female: 55.4% with mild disease vs. 36.1% with severe disease, p = 0.053). The patients who suffered from severe disease were significantly older and had a higher BMI compared to patients with mild disease (61.81±11.25 vs. 47.13±15.95 years, p<0.001 and 29.9±6.81 vs. 26.91±4.985 kg/m$^2$, p = 0.01; respectively). The time between the onset of symptoms and arrival at the emergency room and the length of hospitalization was significantly longer in patients who had severe disease than in those with mild disease (7.17±4.90 vs. 4.38±3.41 days, p = 0.04 and 15.92± 8.42 vs. 2.54±3.42 days, p<0.0001; respectively). Hospitalization duration was significantly longer in patients with high risk for OSA compared to those with low risk for OSA (10.97±9.43 vs. 4.71±6.86 days, respectively; p = 0.001) (Table 2).

Regression analyses were performed to further examine the association between OSA risk and COVID-19 severity. After adjusting for BMI, age, hypertension, and chronic disease, the odds ratio was 4.3 (95% CI 1.2–16, p = 0.029).

**Table 1. Characteristics of the COVID-19 patients.**

| Characteristics | N (%) |
|---|---|
| Age (years), M (SD) | 51.58 (16.3) |
| BMI (kg/m$^2$), M (SD) | 27.85 (5.73) |
| Sex (male) | 60 (50.4) |
| COVID-19 (severe) | 36 (30.3) |
| Days of admission, M (SD) (range) | 6.62 (8.22) |
| Smoking history | 3 (2.5) |
| Hypertension | 25 (21.0) |
| Type 2 diabetes | 20 (18.2) |
| Pulmonary disease | 11 (9.3) |
| Cardiovascular disease | 10 (8.5) |
| Kidney disease | 3 (2.5) |
| Liver disease | 1 (0.8) |
| Other chronic disease | 47 (40.2) |
| Regularly medication consumption | 58 (48.7) |
| Snoring | 34 (51.5) |
| Risk of OSA | 37 (31.1) |
| Sleeping hours M (SD) | 6.22 (1.51) |
| ESS (hours) M (SD) | 7.60 (5.37) |

BMI—body mass index, OSA—obstructive sleep apnea, ESS—Epworth sleeping scale

COVID-19—coronavirus disease 2019, M—mean, SD—standard deviation

**Table 2. Association between the various characteristics of COVID-19 patients with the severity of the disease.**

| Characteristics \| COVID-19 severity | Mild (n = 83) | Severe (n = 36) | p-value |
|---|---|---|---|
| Age (years), M (SD) | 47.14 (15.95) | 61.81 (11.25) | .000 |
| BMI (kg/m$^2$), M (SD) | 26.91 (4.95) | 29.94 (6.81) | .010 |
| Sex (male) | 37 (44.6) | 23 (63.9) | .083 |
| Interval between the onset of symptoms and arrival at the emergency room (days) M (SD) | 4.38 (3.41) | 7.17 (4.90) | .004 |
| Days of admission, M (SD) | 2.54 (3.41) | 15.92 (8.42) | .000 |
| Smoking history | 3 (3.6) | 0 (0.0) | .553 |
| Hypertension | 18 (22.5) | 7 (21.2) | 1.00 |
| Type 2 diabetes | 12 (15.8) | 8 (23.5) | .481 |
| Pulmonary disease | 9 (11.0) | 2 (5.6) | .556 |
| Cardiovascular disease | 7 (8.5) | 3 (8.3) | 1.00 |
| Kidney disease | 2 (2.4) | 1 (2.8) | 1.00 |
| Liver disease | 1 (1.2) | 0 (0.0) | 1.00 |
| Snoring | 24 (53.3) | 10 (47.6) | .866 |
| Risk of OSA | 19 (22.9) | 18 (50.0) | .007 |
| Sleeping hours M (SD) | 6.10 (1.21) | 6.54 (2.10) | .327 |
| ESS (hours) M (SD) | 8.09 (5.22) | 6.45 (5.62) | .143 |

BMI—body mass index, OSA—obstructive sleep apnea, ESS—Epworth sleeping scale

COVID-19—coronavirus 2019, M—mean, SD—standard deviation

In addition, the patients with high risk for OSA were significantly older (57.07±10.94 vs. 49.0±17.49 years, p = 0.003), had a higher BMI (31.61±6.19 vs. 31.61±6.19 kg/m$^2$, p<0.001) and were mostly males (70.3% vs. 41.5%, p = 0.007) compared to the low-risk patients. Hypertension and snoring were significantly more common in the high-risk group than in the low-risk group (44.1% vs. 12.7%; p<0.0001 for hypertension; 81.8% vs. 36.4%, p = 0.01 for snoring) (Table 3).

**Table 3. Association between the various characteristics of COVID-19 patients and the risk for OSA.**

| Characteristics \| Risk of developing OSA | Low (n = 82) | High (n = 37) | p-value |
|---|---|---|---|
| Age (years), M (SD) | 49.10 (17.49) | 57.07 (10.94) | .003 |
| BMI (kg/m$^2$), M (SD) | 26.16 (4.65) | 31.61 (6.19) | .000 |
| Sex (male) | 34 (41.5) | 26 (70.3) | .007 |
| Covid-19 (severe) | 18 (22.0) | 18 (48.6) | .007 |
| Days of admission, M (SD) | 4.71 (6.86) | 10.97 (9.43) | .001 |
| Smoking history | 2 (2.4) | 1 (2.7) | 1.00 |
| Hypertension | 10 (12.7) | 15 (44.1) | .001 |
| Type 2 diabetes | 10 (13.2) | 10 (29.4) | .076 |
| Pulmonary disease | 5 (6.2) | 6 (16.2) | .162 |
| Cardiovascular disease | 5 (6.2) | 5 (13.5) | .331 |
| Kidney disease | 2 (2.5) | 1 (2.7) | 1.00 |
| Liver disease | 1 (1.2) | 0 (0.0) | 1.00 |
| Snoring | 16 (36.4) | 18 (81.8) | .001 |

BMI—body mass index, OSA—obstructive sleep apnea, COVID-19—coronavirus disease 2019, M—mean, SD—standard deviation

There was no significant difference between patients with severe disease and those with mild disease with regard to various comorbidities, regular medication consumption, or snoring. The percentage of patients with severe disease at risk for OSA was significantly higher than that of patients with mild disease (50% vs. 22.9%, p = 0.003).

The Berlin questionnaire was used to evaluate the risk of OSA. Of the 119 patients included in the study, 37 (31.1%) were at risk of OSA, although they had not been previously diagnosed with OSA.

The data revealed that a significant percentage of patients who demonstrated a high risk for OSA suffered from severe COVID-19 compared to patients with a low risk for OSA (48.6% vs. 22%, respectively, p = 0.007) (Table 3).

## Discussion

Although the COVID-19 pandemic broke out only about 2 years ago, there is already well-founded evidence that OSA is a risk factor for worse prognosis, longer hospitalization, and higher risk of death [14, 16, 17]. In addition, a higher prevalence of OSA was observed in patients hospitalized with COVID-19 [14]. We screened patients diagnosed with COVID-19, hospitalized, or admitted to a community hotel setting. Our study results revealed that OSA is a risk factor for severe COVID-19 and, consequently, a long duration of hospitalization. Moreover, patients with a high risk for OSA were significantly older with higher BMI, mostly males, and presented higher rates of hypertension and snoring than those with a low risk for OSA.

Similar to our study, Iannella et al. used the STOP-BANG questionnaire to investigate the association between OSA risk in infected patients and disease severity. The questionnaire was completed by the COVID-19 patients on admission. Their study revealed that 41.6% of the patients who required enhanced respiratory support demonstrated a high risk of OSA. Only 20.8% of the hospitalized patients who received conventional oxygen therapy presented a high risk for OSA (p = 0.05) [26]. Similarly, one study included 20,598 anonymous volunteers, and 9.5% were observed to be at high risk for OSA, according to the STOP-BANG questionnaire. A high risk for OSA and diabetes is associated with the COVID-19 diagnosis. Moreover, a high risk for OSA, male sex, diabetes, and depression was associated with an increased risk of hospitalization or intensive care unit (ICU) treatment [27].

Risk for OSA influences the severity of the acute disease and recovery. Peker et al. [15] reported the clinical outcome of COVID-19 patients who presented high- or low-risk for OSA based on the Berlin questionnaires administered to the patients in the outpatient clinic shortly after discharge from the hospital. Clinical improvement within two weeks was reported in 75.4% of 242 patients (out of 320) hospitalized in the high-risk OSA group and 88.4% in the low-risk OSA group (p = 0.014). In the multivariate regression analyses, high-risk OSA and male sex were observed to be factors predicting a delay in clinical improvement. In the entire study population, a high risk of OSA was associated with clinical deterioration and a greater need for oxygen support.

Many studies have demonstrated the effect of OSA on the severity of COVID-19 symptoms [28, 29]. Voncken et al. compared the impact of OSA on the clinical outcomes of COVID-19 and observed that mortality and duration of hospitalization were increased in COVID-19 patients with OSA [29]. In a study that screened 9,405 COVID-19 patients, OSA was observed to be more prevalent in patients requiring hospitalization than in those who did not (15.3% vs. 3.4%, p<0.0001) and in those who demonstrated respiratory failure (19.4% vs. 4.5%, p<0.0001) [14]. A meta-analysis of 21 studies with 54,276 COVID-19 patients showed that OSA was associated with poor outcomes, severe COVID-19, ICU admissions, the need for mechanical ventilation, and mortality [11].

Several pathophysiological mechanisms associated with OSA may contribute to the aggravation of COVID-19. The renin-angiotensin system (RAS), a hormone system that regulates blood pressure, was found to be dysregulated in OSA and observed to regulate the expression of the entry receptor of SARS-CoV-2 and the angiotensin-converting enzyme [30–34]. Consequently, RAS raises the viral susceptibility of OSA patients, thereby contributing to the risk of severe COVID-19. A meta-analysis of 13 studies revealed that higher levels of angiotensin II and aldosterone were observed in patients with OSA [34].

Breathing disruptions during sleep that accompany OSA are associated with intermittent blood gas disturbances (hypercapnia and hypoxemia) and stimulation of sympathetic activation, which may affect the respiratory function of patients with COVID-19 [25]. Inflammation, a common condition among patients with OSA, is associated with increased levels of proinflammatory substances such as ferritin, interleukin-6 and 17, leptin, and tumor necrosis factor. Tumor necrosis may contribute to the increase in hypoxemia and cytokine storm, leading to multiorgan failure in patients with COVID-19 pneumonia [35–38]. In addition, OSA is characterized by resistance to the anti-inflammatory effects of corticosteroids and potentially contributes to the effect of OSA on COVID-19 pathogenesis [35, 39]. Ho et al. examined the association between some factors that affect OSA severity and COVID-19. They revealed that low oxyhemoglobin desaturation, which often occurs in patients with OSA, was significantly associated with COVID-19 severity in hospitalized or ICU patients [39].

Aspiration during sleep is one of the pathways through which snoring can increase vulnerability to pneumonia. The vibration of the pharyngeal airway during snoring allows the sucking of large amounts of virus-loaded saliva and mucus from the nasopharyngeal airway into the lower airways and lungs [22]. Oral lung aspiration contributes to many lower airway infectious diseases [40]. Thus, it might be suggested that nasal susceptibility to COVID-19 and subsequent aspiration plays a key role in mediating virus seeding into the lungs. Wölfel et al. analyzed virus replication in tissues of the upper respiratory tract in nine COVID-19 cases. They revealed that early infection in the upper respiratory tract (0–5 days) was followed by subsequent aspiration and infection of the lower lung [41].

## Limitations

Our study had some limitations. First, this retrospective observational study included a small sample of patients from a single medical center. In addition, patients with very severe diseases who died were not included in the study; thus, we could not obtain data regarding their risk for OSA. The data from these patients could have further contributed to the findings of this study. In addition, patients filled out the Berlin questionnaire retrospectively after developing COVID-19. This could have led to a bias, mainly in the fatigue-sleepiness section.

Second, the study was conducted in the first to third wave of the COVID-19 pandemic; thus, our findings may not apply to the Omicron era.

Third, we do not have long-term follow-up data about the patients, so we could not assess their long-term outcomes or confirm whether they had obstructive sleep apnea by completing a sleep test.

## Conclusion

Several studies have presented evidence that OSA can be considered a risk factor for severe COVID-19. Our study findings demonstrated that OSA was associated with severe COVID-19 and longer hospitalization. Although OSA is very common, many patients are underdiagnosed. Thus, we recommend that clinicians be aware that some COVID-19 patients who have not been diagnosed with OSA could still be at high risk for OSA, thus making them prone to

severe COVID-19. A routine mandatory request to fill the Berlin and ESS questionnaires for every hospitalized COVID-19 patient, especially those with comorbidities, could allow for closer monitoring of these patients, and possibly early treatment with continuous positive airway pressure may prevent the deterioration to a more serious illness.

## Author Contributions

**Conceptualization:** Nissim Arish, Shmuel Goldberg.

**Data curation:** Nissim Arish.

**Investigation:** Nissim Arish, Gabriel Izbicki, Ariel Rokach, Amir Jarjou'i, George Kalak.

**Supervision:** Nissim Arish, Shmuel Goldberg.

**Validation:** Nissim Arish, Shmuel Goldberg.

**Writing – original draft:** Nissim Arish.

**Writing – review & editing:** Nissim Arish, Gabriel Izbicki, Ariel Rokach, Shmuel Goldberg.

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
