## [Decision Letter · Decision Letter 0]

16 Jan 2023

PONE-D-22-31274

Association of the risk of obstructive sleep apnoea with the severity of COVID-19

PLOS ONE

Dear Dr. Arish,

Thank you for submitting your manuscript to PLOS ONE. After careful consideration, we feel that it has merit but does not fully meet PLOS ONE’s publication criteria as it currently stands. Therefore, we invite you to submit a revised version of the manuscript that addresses the points raised during the review process.

The topic is interesting. However some issues can be raised: 

a) as stated by the Authors, two different COVID waves were considered. Were there any differences? and what about vaccination?

b) the risk of OSA was assessed. We suggest to delete paragraphs describing OSA and its physiopatology since no enrolled patient had a diagnosis of OSA?

c) we suggest to discuss to limits (if any) of the assessment of the "risk" of OSA

Follow-up data, if available, should be added. If not available, this shoud be addressed as a potential limitation 

We look forward to receiving your revised manuscript.

Kind regards,

Chiara Lazzeri

Academic Editor

PLOS ONE

Journal Requirements:

3. We note that the "Berlin Questionnaire© Sleep Apnea” in your submission contain copyrighted images. All PLOS content is published under the Creative Commons Attribution License (CC BY 4.0), which means that the manuscript, images, and Supporting Information files will be freely available online, and any third party is permitted to access, download, copy, distribute, and use these materials in any way, even commercially, with proper attribution. For more information, see our copyright guidelines: http://journals.plos.org/plosone/s/licenses-and-copyright.

a. You may seek permission from the original copyright holder of "Berlin Questionnaire© Sleep Apnea” to publish the content specifically under the CC BY 4.0 license.

“This study received no funding.”

Reviewers' comments:

Reviewer's Responses to Questions

**Comments to the Author**

1. Is the manuscript technically sound, and do the data support the conclusions?

Reviewer #1: Yes

2. Has the statistical analysis been performed appropriately and rigorously? 

Reviewer #1: Yes

3. Have the authors made all data underlying the findings in their manuscript fully available?

Reviewer #1: Yes

4. Is the manuscript presented in an intelligible fashion and written in standard English?

Reviewer #1: Yes

5. Review Comments to the Author

Reviewer #1: In this paper the authors examine the association between risk for OSA and severe COVID-19. The paper is clearly written and the authors describe the limitations clearly in the limitations section. The authors should elucidate the following major points:

1. There is no mention of the data collection time that I would have found. It is important to describe for the patients, which wave of pandemic wave the data comes from. a) Were the individuals eligible for vaccination,b) if they were, how did vaccination status associate with disease outcome, and finally c) did they receive Paxlovid during acute infection.

2. Since the study is retrospective, is there information on long-term symptoms like post-acute COVID-19 syndrome / Long COVID in the individuals and did that affect association with fatigue?

3. Did the patients receive a formal diagnosis of OSA during follow-up?

4. Representation of P-values with notion less than 0.001 would be easier to read than exact values of P = 0.00.

5. Table 3 is titled “Risk to develop OSA” but shouldn’t it be “Individuals at risk for developing OSA”

6. Perhaps it would make sense to discuss the high proportion of severe cases of COVID-19. Is it related to the recruitment through a clinic, sociodemographic factors or risk profile of the patients?

6. PLOS authors have the option to publish the peer review history of their article (what does this mean?). If published, this will include your full peer review and any attached files.

Reviewer #1: No

---

## [Author Response · Author response to Decision Letter 0]

12 Mar 2023

Thank you for the opportunity to resubmit our revised manuscript. We have carefully read the reviewers' remarks and suggestions and have modified the manuscript accordingly. The changes in the revised manuscript are in red font. 

We have responded to all comments and suggestions in a point-by-point manner immediately below this letter, where the reviewers' comments are in black and our responses are in red. 

“The authors received no specific funding for this work.” The statistical analysis is available in OSF repository

DOI 10.17605/OSF.IO/7VUTX

We hope you will find our responses satisfactory. We believe that our research shows novel findings, and we look forward to working with you and the reviewers to move this manuscript closer to publication in PLOS ONE.

We would be glad to respond to any further questions and comments you may have, and we look forward to hearing from you.

---

## [Editor Report · Decision Letter 1]

23 Mar 2023

Association of the risk of obstructive sleep apnoea with the severity of COVID-19

PONE-D-22-31274R1

Dear Dr. Arish,

We’re pleased to inform you that your manuscript has been judged scientifically suitable for publication and will be formally accepted for publication once it meets all outstanding technical requirements.

Kind regards,

Chiara Lazzeri

Academic Editor

PLOS ONE
---

## [Editor Report · Acceptance letter]

29 Mar 2023

PONE-D-22-31274R1 

Association of the risk of obstructive sleep apnoea with the severity of COVID-19 

Dear Dr. Arish:

I'm pleased to inform you that your manuscript has been deemed suitable for publication in PLOS ONE. Congratulations! Your manuscript is now with our production department. 

Kind regards, 

on behalf of

Dr. Chiara Lazzeri 

Academic Editor

PLOS ONE